Accumulation of di-2-ethylhexyl phthalate from polyvinyl chloride flooring into settled house dust and the effect on the bacterial community

Velazquez Samantha 1
http://orcid.org/0000-0003-0131-0310 Bi Chenyang 2 3
Kline Jeff 1 4
Nunez Susie 1
Corsi Rich 3 5
Xu Ying 3 6
http://orcid.org/0000-0002-2615-8055 Ishaq Suzanne L. 1 7 sue.ishaq@maine.edu
1 Biology and the Built Environment Center, University of Oregon , Eugene, OR , USA
2 Department of Civil Engineering, Virginia Polytechnic Institute and State University (Virginia Tech) , Blacksburg, VA , USA
3 Department of Civil, Architectural and Environmental Engineering, University of Texas at Austin , Austin, TX , USA
4 Energy Studies and Buildings Laboratory, University of Oregon , Eugene, OR , USA
5 Fariborz Maseeh College of Engineering and Computer Science, Portland State University , Portland, OR , USA
6 Department of Building Science, Tsinghua University (清华大学化学系) , Beijing , China
7 School of Food and Agriculture, University of Maine , Orono, ME , USA
Mortimer Monika
Electronic publication date: 2019 Nov 22
Publication date: 2019
Volume: 7
Electronic Location ID: e8147
Received 2019 Sep 24; Accepted 2019 Nov 3
Copyright: © 2019 Velazquez et al.
Copyright year: 2019
Copyright holder: Velazquez et al.
License: This is an open access article distributed under the terms of the Creative Commons Attribution License, which permits unrestricted use, distribution, reproduction and adaptation in any medium and for any purpose provided that it is properly attributed. For attribution, the original author(s), title, publication source (PeerJ) and either DOI or URL of the article must be cited.
License URL: https://creativecommons.org/licenses/by/4.0/

Keywords: DEHP, Gas chromotography, Illumina MiSeq, Indoor microbiome, Vinyl flooring

Funding: Alfred P. Sloan Foundation to the Biology and the Built Environment Center (BioBE) at the University of Oregon This work was funded by grants from the Alfred P. Sloan Foundation to the Biology and the Built Environment Center (BioBE) at the University of Oregon. The funders had no role in study design, data collection and analysis, decision to publish, or preparation of the manuscript.

==============================
Di-2-ethylhexyl phthalate (DEHP) is a plasticizer used in consumer products and building materials, including polyvinyl chloride flooring material. DEHP adsorbs from material and leaches into soil, water, or dust and presents an exposure risk to building occupants by inhalation, ingestion, or absorption. A number of bacterial isolates are demonstrated to degrade DEHP in culture, but bacteria may be susceptible to it as well, thus this study examined the relation of DEHP to bacterial communities in dust. Polyvinyl chloride flooring was seeded with homogenized house dust and incubated for up to 14 days, and bacterial communities in dust were identified at days 1, 7, and 14 using the V3–V4 regions of the bacterial 16S rRNA gene. DEHP concentration in dust increased over time, as expected, and bacterial richness and Shannon diversity were negatively correlated with DEHP concentration. Some sequence variants of Bacillus, Corynebacterium jeddahense, Streptococcus, and Peptoniphilus were relatively more abundant at low concentrations of DEHP, while some Sphingomonas, Chryseobacterium, and a member of the Enterobacteriaceae family were relatively more abundant at higher concentrations. The built environment is known to host lower microbial diversity and biomass than natural environments, and DEHP or other chemicals indoors may contribute to this paucity.

Introduction

Di-2-ethylhexyl phthalate (DEHP), a known endocrine-disrupting chemical, is widely used in consumer goods and building materials, from where it is leached into dust, water, or soil and is readily inhaled, absorbed, or ingested by building occupants (Xu et al., 2009; Just et al., 2015; Rowdhwal & Chen, 2018; Bi et al., 2018). DEHP concentration is significantly higher in settled (i.e., surface) dust in homes with polyvinyl flooring (Just et al., 2015; Bi et al., 2018), as well as in HVAC filters (Bi et al., 2018), indicating regular aerial dispersion from floor surfaces. DEHP has been associated with increased reports of children’s asthma during warmer seasons (Bi et al., 2018) and body mass index (Wang et al., 2013), and its byproducts or metabolites can likewise be hazardous to human health (Just et al., 2015; Rowdhwal & Chen, 2018; Bi et al., 2018; Bope et al., 2019).

Due to the proximity to human occupants and risk for exposure, indoor chemistry is relevant to health (Just et al., 2015; Wang et al., 2017; Rowdhwal & Chen, 2018). But our understanding of indoor chemistry is complicated by the diversity of chemical compounds present in the built environment, emission from materials, chemical interactions (e.g., adsorption, absorption) with materials, and local environmental conditions such as ventilation rate and temperature, all of which can alter chemical reactivity (Bi, Liang & Xu, 2015; Weschler & Carslaw, 2018). Moreover, some monocultures of bacteria may degrade chemical contaminants in vitro (Chao & Cheng, 2007; Vamsee-Krishna & Phale, 2008; Liang et al., 2008; Yang et al., 2018), including the phthalates that are emitted from vinyl flooring, as well as in soil or other environmental samples (Liang et al., 2008; Yang et al., 2018; Zhu et al., 2018). However, in situ microbial communities in indoor dust have only been demonstrated to degrade chemical contaminants under conditions of high relative humidity (80–100%), and this appeared to be largely due to fungal degradation (Bope et al., 2019). It has been suggested that microorganisms naturally produce DEHP, particularly microorganisms in soils (Ortiz & Sansinenea, 2018). Regardless, the presence of DEHP or other chemicals in dust may affect microbial survival (Brundrett et al., 2018; Lotfy et al., 2018; Zhu et al., 2018), which would alter the composition of microorganisms in dust and potential community dynamics.

Little research has been done to elucidate the relationships between phthalate composition and the effects on microbial communities in dust. This pilot study investigated whether dust which accumulated DEHP emitted from polyvinyl chloride flooring would affect the bacterial communities in dust gathered from homes.

Materials and Methods

Experimental design was similar to previous studies (Wu et al., 2016; Bi et al., 2018). Polyvinyl chloride flooring was cut into 60 mm diameter specimens, placed into glass petri dishes, and sprayed evenly with freshly collected, homogenized house dust containing living microbial communities (Fahimipour et al., 2018) at a concentration of 5 g/m2. The dishes were placed into a 10 L stainless steel container with one inlet and one outlet for airflow, with an air change rate of 0.5 h−1, ∼0% relative humidity (RH), 25 ± 0.5 °C and no light. Experimental control samples of dust in glass petri dishes, one per time point, were subjected to identical conditions without the inclusion of vinyl flooring, however the homogenized house dust did contain some background DEHP (291 μg/g dust). All samples were randomly placed in the steel container and selected for removal at each time period. During the incubation periods, DEHP was passively adsorbed from polyvinyl flooring and loaded into dust in the experimental samples.

Petri dishes were cleaned three times with hexane through ultrasonication and baked at 80 °C for 12 h before experiments. Prior to sample measurements, five matrix spiking experiments were conducted for DEHP; and the recovery efficiencies were between 72% and 102%. Additionally, all samples were spiked with three surrogate chemicals (i.e., dibenzyl phthalate, diphenyl isophthalate and diphenyl phthalate) to determine analytical recovery efficiencies. The average recovery for the surrogates in all samples was 85.3 ± 7.3%. Each sample was weighed three times using a five-digit microbalance and mean weight was calculated. To check drift of the GC system, calibration standards were run after every six samples, and results were accepted only when the standards varied less than 10%. A calibration curve, which has a R2 > 0.9999, was constructed using six different concentrations in the range of 1–500 ng/µL for DEHP. Lab blanks (i.e., empty petri dish) were collected and treated identically to the samples (except that no dust was collected). None of the target compounds were detected in these laboratory blanks. To avoid contamination, all glassware and metal ware used for sample collection and analysis were ultrasonically cleaned in methanol for 20 min, rinsed with methanol three times, and then baked at 80 °C in an oven prior to use.

Following the incubation period, the dust placed on the source material within the petri dish was removed gently with a quantitative filter and transferred to an eight ml glass vial. The vial was weighed before and after placing the dust to determine the mass of dust collected in each petri dish. The collected dust samples were ultrasonically extracted three times for 30 min each with n-hexane (Hexane (mixture of isomers) ≥98.5% ACS, VWR Chemicals BDH). The volume of the extract was concentrated to approximately 500 µL by gently bubbling the samples with ultrapure nitrogen. DEHP concentrations in the extract was then analyzed using a GC-FID system (Agilent 7890A) equipped with a DB-5ht column (30 m, 0.25 mm, 0.1 µm). The GC-FID system was operated using a splitless injection at 275 °C. The temperature program was set at 120 °C and held for 2 min, ramped up 12 °C/min for 15 min and held 3 min, and then ramped up 20 °C/min for 2 min and held 2 min. The detector was set at 320 °C. Helium was used as the carrier gas with a constant column flow set at 1.2 ml/min. A calibration standard was run prior to each GC injection, and the variance was always below 10% for all injections.

Sterile cotton swabs were used to collect dust from remaining petri dishes for microbial analysis. A dust sample was collected at time point 0, and three experimental replicates were collected each at day 1, 7, and 14, using sterile swabs. A sterile swab and petri dish, and no-template extraction and PCR samples, were included as negative controls. Plates were stored at −20 °C until DNA extraction was performed; swab tips were added directly to bead tubes for extraction. Nucleic acids were extracted using the MoBio PowerSoil DNA Extraction Kit (MoBio, Carlsbad, CA, USA) following kit instructions.

The V3 and V4 (319F-806R) regions of the 16S rRNA gene were PCR-amplified following a previously described protocol (Kembel et al., 2014), and amplicons were purified with a bead-based clean-up using Mag-Bind RxnPure Plus (Omega Bio-tek, Norcross, GA, USA). Cleaned DNA was quantified using Quant-iT dsDNA assay kit, and pooled with equal concentrations of amplicons for Illumina Miseq ver. 4 paired-end sequencing using a 250-cycle kit at the University of Oregon Genomics Core (Eugene, OR, USA). Raw data are available through the National Center for Biotechnology Information (NCBI) Sequence Read Archive (SRA) under BioProject accession PRJNA541585.

Sequence data were processed using the DADA2 pipeline (Callahan, McMurdie & Holmes; Callahan et al., 2016) in R statistical program (R Core Team, 2018). During filtering, sequences were trimmed by 10 bases at the start position, and 10 and 20 bases from the end position for forward and reverse reads, respectively. No ambiguous bases were permitted, and the maximum expected errors were two and three for forward and reverse, respectively. NA matching the phiX sequencing controls was removed. Error rates were learned on 1 × 106 randomly-selected dereplicated reads, and used to identify sequence variants (SVs), after which forward and reverse reads were merged into contiguous sequences. Bimeras (two-parent chimeras) were identified and removed, and taxonomy assigned using the Silva NR ver 132 database (Yilmaz et al., 2014). Laboratory negative controls were used to remove identical SVs from experimental samples, using adapted code (Ishaq, 2017). Two experimental controls did not successfully generate sequences, and were included in certain figures but removed from the analysis prior to statistical comparison. Samples were rarefied to 6,732 sequences per sample, and statistical comparison was performed with R packages phyloseq (McMurdie & Holmes, 2013), vegan (Oksanen, 2018), and rfPermute (Breiman et al., 2015; Archer, 2019).

Results and discussion

Home dust was incubated on polyvinyl chloride flooring, from which the dust was seeded with di-2-ethylhexyl phthalate (DEHP) over 14 days, averaging 1,041 μg/g on day 1, 4,547 μg/g on day 7, and 10,066 μg/g on day 14 (Fig. S1).

Bacterial communities in dust samples exhibited reduced richness on day 14 (Fig. 1A; lm, t = −2.560, p = 0.043, adjusted R2 = 0.366) and was negatively correlated with DEHP concentration, although significance was only trending (lm, t = −1.939, p = 0.094, adjusted R2 = 0.256). Shannon diversity was not affected by Day or DEHP concentration (Fig. 1B; Mann–Whitney, p > 0.05). Control dust, which was incubated on glass plates, exhibited variable richness and diversity, but days 0 and 7 had such low richness (Fig. S2) as to be removed during rarefaction. As no replicates were included in microbial analysis it is unclear whether sample days 0 and 7 failed sequencing procedures or accurately had lower bacterial richness.

Figure 1 Observed richness (SVs) and Shannon diversity for bacteria in dust, which was incubated on polyvinyl chloride flooring and thereby accumulated di-2-ethylhexyl phthalate, over 14 days.

Data are rarefied and quality-filtered.

Dust samples clustered significantly by Day Fig. 2; PermANOVA, unweighted Jaccard, f = 1.147, R2 = 0.141, p = 0.016. A number of bacterial taxa were associated with DEHP concentration (Fig. 3). In particular, two Bacillus sp., Corynebacterium jeddahense, a Streptococcus sp., and a Peptoniphilus sp. were relatively more abundant at low concentrations, while multiple Sphingomonas sp, a Chryseobacterium sp., and a member of the Enterobacteriaceae family were relatively more abundant at higher concentrations of DEHP. Several bacterial taxa were significantly associated with control samples with no DEHP (Fig. 3B), only some of which showed a decline in relative abundance between days 1 and 14. Phylum-level taxonomy for all samples is provided in Fig. S3.

Figure 2 Non-metric multidimensional scaling plot of bacterial community membership in dust, which was incubated on polyvinyl chloride flooring and thereby accumulated di-2-ethylhexyl phthalate, over 14 days.

Lowest stress = 0.067.

Figure 3 Bacterial taxa significantly associated with di-2-ethylhexyl phthalate in dust (A), compared to controls (B).

Panels are faceted by Day (1, 7, 14). Only significant features are shown (rfpermute, p < 0.05).

Bacterial community richness declined over the 14 day study period, and while only trending towards being affected by DEHP, this may reflect low statistical power. And while DEHP concentration is inherently linked to Day, we speculate that the marginal effect observed here is due to SVOC concentration. Bacteria are known to survive in dust even after 90 days with no input of resources, and in particular, when incubated in the dark (Fahimipour et al., 2018). In the present study, only two controls had sufficient data for analysis, but the day 14 sample did have higher richness than the control day 1. Additionally, the bacterial taxa which were significantly associated with the controls showed similar relative abundance between the two time points. We posit that the bacterial community in our control samples remained stable during the experiment.

However, a more thorough investigation is needed to understand the dynamic here, including additional time points, replication in the controls, and culture assays to determine if bacteria which changed in relative abundance indeed increase or decrease when incubated with DEHP. It is also possible that DEHP itself had no direct effect, but that the degradation or utilization of DEHP by certain taxa created byproducts or competitive inhibitors which affected the microbial community. While preliminary, these results represent the first investigation into the effect of DEHP on an entire bacterial community in dust. Despite the recent scientific advances in indoor chemistry and indoor microbiology, respectively, very little work has been done to elucidate the interaction between chemicals in the built environment and the microorganisms found there.

Semi-volatile organic compounds (SVOCs) are commonly incorporated into manufactured products for a variety of purposes, but in building materials they may leach and accumulate in dust on surfaces, adsorb to other materials, or react to form a variety of chemical byproducts (Lucattini et al., 2018; Bi et al., 2018). DEHP readily adsorbs and desorbs from suspended particles, that is, dust, and can be readily transported this way (Xu et al., 2009). Moreover, in addition to facilitating particle transport, increasing air exchange only increases DEHP emission rates from vinyl flooring (Xu et al., 2009). Thus, DEHP could potentially affect bacterial communities not adjacent to vinyl sources. The built environment is known to host lower microbial diversity and biomass than natural environments (Gibbons, 2016; NASEM, 2017), and the presence of DEHP or other SVOCs may be contributing to this paucity.

Supplemental Information

Supplemental Information 1 Di-2-ethylhexyl phthalate concentration in experimental samples of dust incubated on polyvinyl chloride floor over 14 days.

Click here for additional data file.

Supplemental Information 2 Observed richness (SVs) and Shannon Diversity for all rarefied, quality-filtered sequence data, for bacterial communities in dust, which was incubated on polyvinyl chloride flooring and thereby accumulated di-2-ethylhexyl phthalate, over 14 days.

Click here for additional data file.

Supplemental Information 3 Phylum-level taxonomy of bacterial communities in dust, which was incubated on polyvinyl chloride flooring and thereby accumulated di-2-ethylhexyl phthalate, over 14 days.

Click here for additional data file.

Additional Information and Declarations

Competing Interests

Author Contributions

Data Availability

The authors declare that they have no competing interests.

Samantha Velazquez analyzed the data, prepared figures and/or tables, authored or reviewed drafts of the paper, approved the final draft.

Chenyang Bi performed the experiments, analyzed the data, authored or reviewed drafts of the paper, approved the final draft.

Jeff Kline conceived and designed the experiments, authored or reviewed drafts of the paper, approved the final draft.

Susie Nunez performed the experiments, authored or reviewed drafts of the paper, approved the final draft.

Rich Corsi conceived and designed the experiments, authored or reviewed drafts of the paper, approved the final draft.

Ying Xu conceived and designed the experiments, contributed reagents/materials/analysis tools, authored or reviewed drafts of the paper, approved the final draft.

Suzanne L. Ishaq conceived and designed the experiments, performed the experiments, analyzed the data, contributed reagents/materials/analysis tools, prepared figures and/or tables, authored or reviewed drafts of the paper, approved the final draft.

The following information was supplied regarding data availability:

The DNA sequences and metadata are available from the NCBI Sequence Read Archive (SRA) under BioProject accession PRJNA541585.

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
