# Peer review of "Accumulation of di-2-ethylhexyl phthalate from polyvinyl chloride flooring into settled house dust and the effect on the bacterial community"

_PeerJ, doi:10.7717/peerj.8147_

## Round 0.1 · original submission · Minor Revisions

Please see the comments of the two reviewers below and address these in the response letter as well as in the manuscript. I look forward to reading the revised manuscript.

·

Basic reporting

The manuscript is well-written in good english and clear.
Literature references are recent and in context with the topic
the figures are well done.

Experimental design

The results are too preliminary because the control is not clear (there are not three duplicates) to have a comparative with the experimental data where the DEPH appears.

Validity of the findings

The results are too preliminary because the control is not clear (there are not three duplicates) to have a comparative with the experimental data where the DEPH appears. Is not clear if the effect is due to the DEPH or to any other factor. For example the population of other bacteria genera appears with DEPH, may be these bacteria are secreting some toxic compounds that inhibit the growth of the others, in fact some bacteria can secrete DEPH.
The authors may be should to have into account these possibilities and not to assign to DEPH the responsability of this inhibition.

Additional comments

One posible solution to known if the DEPH was the responsable of the decrement of the bacteria could be to put some DEPH concentration in vitro with different genera of bacteria that authors identificated in dust.
Otherwise is better to give some other posible explanations to explain the inhibition of some bacteria (antagonistic effect, or secreted secondary metabolites effect)

Reviewer 2 ·

Basic reporting

The font in this paper should be uniform.
Page 9, line 133: 20C should be changed to 20 ℃

Experimental design

I would like more assurance as to quality of DEHP data in this study as the author did not give any information about the method blanks, recoveries and reproducible of matrix spiked samples.
Page 9, line 127-128: Not only the variance should be always below 10% for all injections, but also the linearity of calibration curves constructed for DEHP should be good (R2 > 0.99) over concentration ranges relevant to those in the samples.
Page 9, line 128: Have the author analyzed the levels of DEHP in procedure blanks to avoid the lab contamination? And have the authors used the internal standard to test the recoveries of the experimants?

Validity of the findings

Overall, this is an interesting paper that reports data relating to accumulation of di-2-ethylhexyl phthalate (DEHP) from polyvinyl chloride flooring to house dust and their effect on the bacterial community. The authors provide only three time points (1, 7, 14 days) to to test it, while more time points are needed for systematic study.

---

## Round 0.2 · accepted · Accept

There are no further comments.